# MoPro: Webly Supervised Learning with Momentum Prototypes

**Junnan Li, Caiming Xiong, Steven C.H. Hoi**
Salesforce Research
{junnan.li,cxiong,shoi}@salesforce.com

## Abstract

We propose a webly-supervised representation learning method that does not suffer from the annotation unscalability of supervised learning, nor the computation unscalability of self-supervised learning. Most existing works on webly-supervised representation learning adopt a vanilla supervised learning method without accounting for the prevalent noise in the training data, whereas most prior methods in learning with label noise are less effective for real-world large-scale noisy data. We propose momentum prototypes (MoPro), a simple contrastive learning method that achieves online label noise correction, out-of-distribution sample removal, and representation learning. MoPro achieves state-of-the-art performance on WebVision, a weakly-labeled noisy dataset. MoPro also shows superior performance when the pretrained model is transferred to down-stream image classification and detection tasks. It outperforms the ImageNet supervised pretrained model by $+10.5$ on 1-shot classification on VOC, and outperforms the best self-supervised pretrained model by $+17.3$ when finetuned on $1\%$ of ImageNet labeled samples. Furthermore, MoPro is more robust to distribution shifts. Code and pretrained models are available at https://github.com/salesforce/MoPro.

## 1 Introduction

Large-scale datasets with human-annotated labels have revolutionized computer vision. Supervised pretraining on ImageNet (Deng et al., 2009) has been the de facto formula of success for almost all state-of-the-art visual perception models. However, it is extremely labor intensive to manually annotate millions of images, which makes it a non-scalable solution. One alternative to reduce annotation cost is self-supervised representation learning, which leverages unlabeled data. However, self-supervised learning methods (Goyal et al., 2019; He et al., 2019; Chen et al., 2020a; Li et al., 2020b) have yet consistently shown superior performance compared to supervised learning, especially when transferred to downstream tasks with limited labels.

With the help of commercial search engines, photo-sharing websites, and social media platforms, there is near-infinite amount of weakly-labeled images available on the web. Several works have exploited the scalable source of web images and demonstrated promising results with webly-supervised representation learning (Mahajan et al., 2018; Sun et al., 2017; Li et al., 2017; Kolesnikov et al., 2020). However, there exists two competing claims on whether weakly-labeled noisy datasets lead to worse generalization performance. One claim argues that the effect of noise can be overpowered by the scale of data, and simply applies standard supervised learning method on web datasets (Mahajan et al., 2018; Sun et al., 2017; Li et al., 2017; Kolesnikov et al., 2020). The other claim argues that deep models can easily memorize noisy labels, resulting in worse generalization (Zhang et al., 2017; Ma et al., 2018). In this paper, we show that both claims are partially true. While increasing the size of data does improve the model's robustness to noise, our method can substantially boost the representation learning performance by addressing noise.

There exists a large body of literature on learning with label noise (Jiang et al., 2018; Han et al., 2018; Guo et al., 2018; Tanaka et al., 2018; Arazo et al., 2019; Li et al., 2020a). However, existing methods have several limitations that make them less effective for webly-supervised representation learning. First, most methods do not consider out-of-distribution (OOD) samples, which is a major

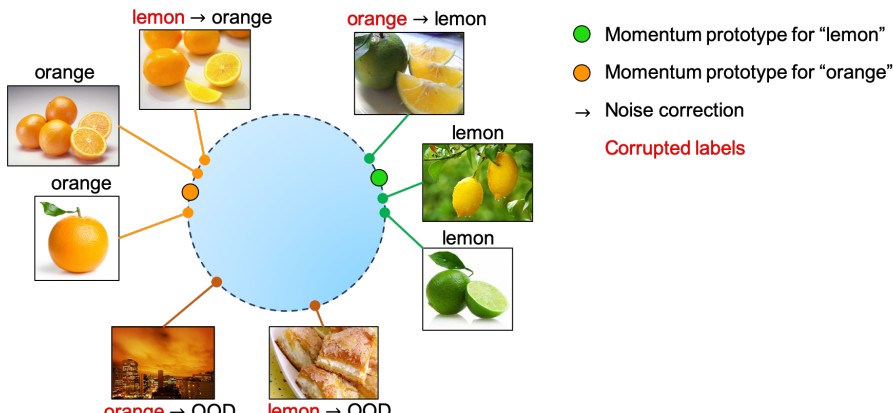

Figure 1: Illustration of the normalized embedding space learned with MoPro. Samples from the same class gather around their class prototype, whereas OOD samples are separated from in-distribution samples. Label correction and OOD removal are achieved based on a sample's distance with the prototypes.

source of noise in real-world web datasets. Second, many methods perform computation-heavy procedures for noise cleaning (Jiang et al., 2018; Li et al., 2019; 2020a), or require access to a set of samples with clean labels (Vahdat, 2017; Veit et al., 2017; Lee et al., 2018), which limit their scalability in practice.

We propose a new method for efficient representation learning from weakly-labeled web images. Our method is inspired by recent developments in contrastive learning for self-supervised learning (He et al., 2019; Chen et al., 2020a; Li et al., 2020b) We introduce *Momentum Prototypes* (MoPro), a simple component which is effective in **label noise correction**, **OOD sample removal**, and **representation learning**. A visual explanation of our method is shown in Figure 1. We use a deep network to project images into normalized low-dimensional embeddings, and calculate the prototype for a class as the moving-average embedding for clean samples in that class. We train the network such that embeddings are pulled closer to their corresponding prototypes, while pushed away from other prototypes. Images with corrupted labels are corrected either as another class or as an OOD sample based on their distance to the momentum prototypes.

We experimentally show that:

- MoPro achieves state-of-the-art performance on the upstream weakly-supervised learning task.
- MoPro substantially improves representation learning performance when the pretrained model is transferred to downstream image classification and object detection tasks. For the first time, we show that weakly-supervised representation learning achieves similar performance as supervised representation learning, under the same data and computation budget. With a larger web dataset, MoPro outperforms ImageNet supervised learning by a large margin.
- MoPro learns a more robust and calibrated model that generalizes better to distribution variations.

## 2 RELATED WORK

### 2.1 WEBLY-SUPERVISED REPRESENTATION LEARNING

A number of prior works exploit large web datasets for visual representation learning (Divvala et al., 2014; Chen & Gupta, 2015; Joulin et al., 2016; Mahajan et al., 2018; Sun et al., 2017; Li et al., 2017; Kolesnikov et al., 2020). These datasets contain a considerable amount of noise. Approximately 20% of the labels in the JMT-300M dataset (Sun et al., 2017) are noisy, whereas 34% of images in the WebVision dataset (Li et al., 2017) are considered outliers. Surprisingly, most prior works have chosen to ignore the noise and applied vanilla supervised method, with the claim that the scale of data can overpower the noise (Mahajan et al., 2018; Sun et al., 2017; Li et al., 2017). However, we show that supervised method cannot fully harvest the power of large-scale weakly-labeled datasets.

Our method achieves substantial improvement by addressing noise, and advances the potential of webly-supervised representation learning.

## 2.2 LEARNING WITH LABEL NOISE

Learning with label noise has been widely studied. Some methods require access to a small set of clean samples (Xiao et al., 2015; Vahdat, 2017; Veit et al., 2017; Lee et al., 2018; Zhang et al., 2020), and other methods assume that no clean labels are available. There exist two major types of approaches. The first type performs label correction using predictions from the network (Reed et al., 2015; Ma et al., 2018; Tanaka et al., 2018; Yi & Wu, 2019; Yang et al., 2020). The second type separates clean samples from corrupted samples, and trains the model on clean samples (Han et al., 2018; Arazo et al., 2019; Jiang et al., 2018; Wang et al., 2018; Chen et al., 2019; Li et al., 2020a). However, existing methods have yet shown promising results for large-scale weakly-supervised representation learning. The main reasons include: (1) most methods do not consider OOD samples, which commonly occur in real-world web datasets; (2) most methods are computational-heavy due to co-training (Han et al., 2018; Li et al., 2020a; Jiang et al., 2018; 2020), iterative training (Tanaka et al., 2018; Yi & Wu, 2019; Wang et al., 2018; Chen et al., 2019), or meta-learning (Li et al., 2019; Zhang et al., 2019).

Different from existing methods, MoPro achieves both label correction and OOD sample removal on-the-fly with a single step, based on the similarity between an image embedding and the momentum prototypes. MoPro also leverages contrastive learning to learn a robust embedding space.

## 2.3 SELF-SUPERVISED REPRESENTATION LEARNING

Self-supervised methods have been proposed for representation learning using unlabeled data. The recent developments in self-supervised representation learning can be attributed to contrastive learning. Most methods (He et al., 2019; Chen et al., 2020a; Oord et al., 2018; Wu et al., 2018) leverage the task of instance discrimination, where augmented crops from the same source image are enforced to have similar embeddings. Prototypical contrastive learning (PCL) (Li et al., 2020b) performs clustering to find prototypical embeddings, and enforces an image embedding to be similar to its assigned prototypes. Different from PCL, we update prototypes on-the-fly in a weakly-supervised setting, where the momentum prototype of a class is the moving average of clean samples' embeddings. Furthermore, we jointly optimize two contrastive losses and a cross-entropy loss.

Current self-supervised representation learning methods are limited in (1) inferior performance in low-shot task adaptation, (2) huge computation cost, and (3) inadequate to harvest larger datasets. We show that weakly-supervised learning with MoPro addresses these limitations.

## 3 METHOD

In this section, we delineate the details of our method. First, we introduce the components in our representation learning framework. Then, we describe the loss functions. Finally, we explain the noise correction procedure for label correction and OOD sample removal. A pseudo-code of MoPro is provided in appendix B.

### 3.1 REPRESENTATION LEARNING FRAMEWORK

Our proposed framework consists of the following components. Figure 2 gives an illustration.

- A noisy training dataset $\{(\boldsymbol{x}_i, y_i)\}_{i=1}^n$, where $\boldsymbol{x}_i$ is an image and $y_i \in \{1, ..., K\}$ is its class label.
- A pseudo-label $\hat{y}_i$ for each image $\boldsymbol{x}_i$, which is its corrected label. Details for generating the pseudo-label is explained in Sec 3.3.
- An encoder network, which maps an augmented image $\tilde{\boldsymbol{x}}_i$ to a representation vector $\boldsymbol{v}_i \in \mathbb{R}^{d_e}$. We experiment with ResNet-50 (He et al., 2016) as the encoder, where the activations of the final global pooling layer ($d_e = 2048$) are used as the representation vector.
- A classifier (a fully-connected layer followed by softmax) which receives the representation $\boldsymbol{v}_i$ as input and outputs class predictions $\boldsymbol{p}_i$.

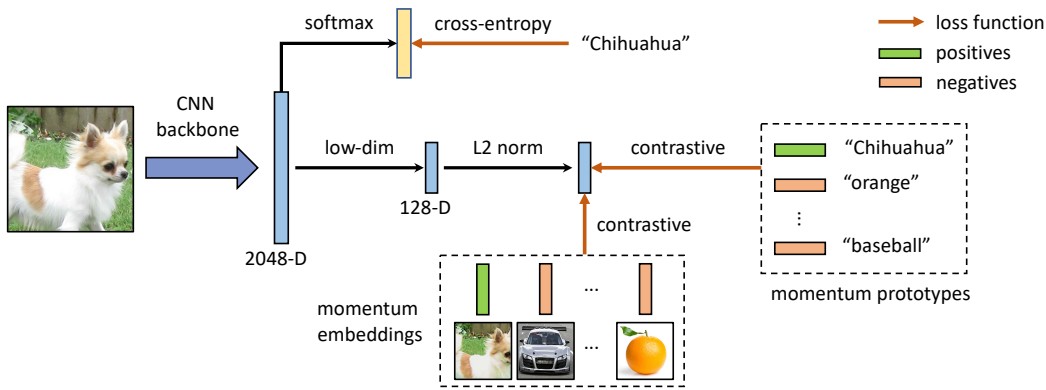

Figure 2: Proposed weakly-supervised learning framework. We jointly optimize a prototypical contrastive loss using momentum prototypes, an instance contrastive loss using momentum embeddings, and a cross-entropy loss using pseudo-labels. The pseudo-label for a sample is generated based on its original training label, the model's prediction, and the sample's distance to the prototypes.

- A projection network, which maps the representation $v_i$ into a low-dimensional embedding $z_i \in \mathbb{R}^{d_p}$ ($d_p = 128$). $z_i$ is always normalized to the unit sphere. Following SimCLR (Chen et al., 2020a), we use a MLP with one hidden layer as the projection network.
- Momentum embeddings $z_i'$ generated by a momentum encoder. The momentum encoder has the same architecture as the encoder followed by the projection network, and its parameters are the moving-average of the encoder's and the projection network's parameters. Same as in MoCo (He et al., 2019), we maintain a queue of momentum embeddings of past samples.
- Momentum prototypes $C \in \mathbb{R}^{d_p \times K}$. The momentum prototype of the $k$-th class, $c_k$, is the normalized moving-average embedding for samples with pseudo-label $\hat{y}_i = k$.

## 3.2 CONTRASTIVE LOSS

As illustrated in Figure 1, we aim to learn an embedding space where samples from the same class gather around its class prototype, while samples from different classes are seperated. We achieve it with two contrastive losses: (1) a prototypical contrastive loss $\mathcal{L}_{\mathrm{pro}}$ which increases the similarity between an embedding and its corresponding class prototype, $(z_i, c_{\hat{y}_i})$, in contrast to other prototypes; (2) an instance contrastive loss $\mathcal{L}_{\mathrm{ins}}$ which increases the similarity between two embeddings of the same source image, $(z_i, z_i')$, in contrast to embeddings of other images. Specifically, the contrastive losses are defined as:

$$\mathcal{L}_{\mathrm{pro}}^i = -\log \frac{\exp(z_i \cdot c_{\hat{y}_i}/\tau)}{\sum_{k=1}^{K} \exp(z_i \cdot c_k/\tau)}, \quad \mathcal{L}_{\mathrm{ins}}^i = -\log \frac{\exp(z_i \cdot z_i'/\tau)}{\sum_{r=0}^{R} \exp(z_i \cdot z_r'/\tau)}, \tag{1}$$

where $\tau$ is a temperature parameter, and $\hat{y}_i$ is the pseudo-label. We use $R$ negative momentum embeddings to construct the denominator of the instance contrastive loss.

We train the classifier with cross-entropy loss, using pseudo-labels as targets.

$$\mathcal{L}_{\mathrm{ce}}^i = -\log(p_i^{\hat{y}_i}) \tag{2}$$

We jointly optimize the contrastive losses and the classification loss. The training objective is:

$$\mathcal{L} = \sum_{i=1}^{n} (\mathcal{L}_{\mathrm{ce}}^i + \lambda_{\mathrm{pro}} \mathcal{L}_{\mathrm{pro}}^i + \lambda_{\mathrm{ins}} \mathcal{L}_{\mathrm{ins}}^i) \tag{3}$$

For simplicity, we set $\lambda_{\mathrm{pro}} = \lambda_{\mathrm{ins}} = 1$ for all experiments.

## 3.3 NOISE CORRECTION

We propose a simple yet effective method for online noise correction during training, which cleans label noise and removes OOD samples. For each sample, we generate a soft pseudo-label $q_i$ by

combining the classifier's output probability $\boldsymbol{p}_i$ with $\boldsymbol{s}_i$, a class probability distribution calculated using the sample's similarity *w.r.t* the momentum prototypes:

$$\boldsymbol{q}_i = \alpha \boldsymbol{p}_i + (1 - \alpha) \boldsymbol{s}_i,$$
$$s_i^k = \frac{\exp(\boldsymbol{z_i} \cdot \boldsymbol{c}_k / \tau)}{\sum_{k=1}^{K} \exp(\boldsymbol{z}_i \cdot \boldsymbol{c}_k / \tau)}. \tag{4}$$

where the combination weight is simply set as $\alpha = 0.5$ in all experiments.

We convert $\boldsymbol{q}_i$ into a hard pseudo-label $\hat{y}_i$ based on the following rules: (1) if the highest score of $\boldsymbol{q}_i$ is above certain threshold $T$, use the class with the highest score as the pseudo-label; (2) otherwise, if the score for the original label $y_i$ is higher than uniform probability, use $y_i$ as the pseudo-label; (3) otherwise, label it as an OOD sample.

$$\hat{y}_i = \begin{cases} \arg\max_k q_i^k & \text{if } \max_k q_i^k > T, \\ y_i & \text{elseif } q_i^{y_i} > 1/K, \\ \text{OOD} & \text{otherwise.} \end{cases} \tag{5}$$

We remove OOD samples from both the cross-entropy loss and the prototypical contrastive loss so that they do not affect class-specific learning, but include them in the instance contrastive loss to further separate them from in-distribution samples. Examples of OOD images and corrected pseudo-labels are shown in the appendices.

### 3.4 Momentum Prototypes

For each class $k$, we calculate its momentum prototype as a moving-average of the normalized embeddings for samples with pseudo-label $k$. Specifically, we update $\boldsymbol{c}_k$ by:

$$\boldsymbol{c}_k \leftarrow \text{Normalize}(m\boldsymbol{c}_k + (1 - m)\boldsymbol{z}_i), \quad \forall i \in \{i \mid \hat{y}_i = k\}, \tag{6}$$

where $\text{Normalize}(\boldsymbol{c}) = \boldsymbol{c}/\|\boldsymbol{c}\|_2$. The momentum coefficient $m$ is set 0.999 in our experiments.

## 4 Experiments

### 4.1 Dataset for upstream training

We use the WebVision (Li et al., 2017) dataset as the noisy training data. It consists of images automatically crawled from Google and Flickr, using visual concepts from ImageNet as queries. We experiment with three versions of WebVision with different sizes: (1) WebVision-V1.0 contains 2.44m images with the same classes as the ImageNet-1k (ILSVRC 2012) dataset; (2) WebVision-V0.5 is a randomly sampled subset of WebVision-V1.0, which contains the same number of images (1.28m) as ImageNet-1k; (3) WebVision-V2.0 contains 16m images with 5k classes.

### 4.2 Implementation details

We follow standard settings for ImageNet training: batch size is 256; total number of epochs is 90; optimizer is SGD with a momentum of 0.9; initial learning rate is 0.1, decayed at 40 and 80 epochs; weight decay is 0.0001. We use ResNet-50 (He et al., 2016) as the encoder. For MoPro-specific hyperparameters, we set $\tau = 0.1, \alpha = 0.5, T = 0.8$ ($T = 0.6$ for WebVision-V2.0). The momentum for both the momentum encoder and momentum prototypes is set as 0.999. The queue to store momentum embeddings has a size of 8192. We apply standard data augmentation (crop and horizontal flip) to the encoder's input, and stronger data augmentation (color changes in MoCo (He et al., 2019)) to the momentum encoder's input. We warm-up the model for 10 epochs by training on all samples with original labels, before applying noise correction.

### 4.3 Upstream task performance

In Table 1, we compare MoPro with existing weakly-supervised learning methods trained on WebVision-V1.0, where MoPro achieves state-of-the-art performance. Since the training dataset

| Method | Architecture | WebVision | | ImageNet | |
|---|---|---|---|---|---|
| | | top-1 | top-5 | top-1 | top-5 |
| Cross-Entropy (Tu et al., 2020) | ResNet-50 | 66.4 | 83.4 | 57.7 | 78.4 |
| MentorNet (Jiang et al., 2018) | InceptionResNet-V2 | 70.8 | 88.0 | 62.5 | 83.0 |
| CurriculumNet (Guo et al., 2018) | Inception-V2 | 72.1 | 89.1 | 64.8 | 84.9 |
| CleanNet (Lee et al., 2018) | ResNet-50 | 70.3 | 87.8 | 63.4 | 84.6 |
| CurriculumNet (Guo et al., 2018; Tu et al., 2020) | ResNet-50 | 70.7 | 88.6 | 62.7 | 83.4 |
| SOM (Tu et al., 2020) | ResNet-50 | 72.2 | 89.5 | 65.0 | 85.1 |
| Distill (Zhang et al., 2020) | ResNet-50 | - | - | 65.8 | 85.8 |
| Cross-Entropy (decoupled) | ResNet-50 | 72.4 | 89.0 | 65.7 | 85.1 |
| MoPro (ours) | ResNet-50 | **73.9** | **90.0** | **67.8** | **87.0** |

Table 1: Comparison with state-of-the-art methods on WebVision-V1.0. Numbers denote accuracy (%) on the clean WebVision-V1.0 validation set and the ILSVRC 2012 validation set. CleanNet (Lee et al., 2018) and Distill (Zhang et al., 2020) require data with clean annotations.

has imbalanced number of samples per-class, inspired by Kang et al. (2020), we perform the following decoupled training steps to re-balance the classifier: (1) pretrain the model with MoPro; (2) perform noise correction on the training data using the pretrained model, following the method in Section 3.3; (3) keep the pretrained encoder fixed and finetune the classifier on the cleaned dataset, using square-root data sampling (Mahajan et al., 2018) which balances the classes. We retrain the classifier for 15 epochs, using a learning rate of 0.01 which is decayed at 5 and 10 epochs. Surprisingly, we also find that a vanilla cross-entropy method with decoupled classifier re-balancing can also achieve competitive performance, outperforming most existing baselines.

## 5 TRANSFER LEARNING

In this section, we transfer weakly-supervised learned models to a variety of downstream tasks. We show that MoPro yields superior performance in image classification, object detection, instance segmentation, and obtains better robustness to domain shifts. Implementation details for the transfer learning experiments are described in appendix C.

### 5.1 LOW-SHOT IMAGE CLASSIFICATION ON FIXED REPRESENTATION

First, we transfer the learned representation to downstream tasks with few training samples. We perform low-shot classification on two datasets: PASCAL VOC2007 (Everingham et al., 2010) for object classification and Places205 (Zhou et al., 2014) for scene recognition. Following the setup by Goyal et al. (2019); Li et al. (2020b), we train linear SVMs using fixed representations from pretrained models. We vary the number $k$ of samples per-class and report the average result

| Method | Pretrain dataset | VOC07 | | | | | Places205 | | | | |
|---|---|---|---|---|---|---|---|---|---|---|---|
| | | $k$=1 | $k$=2 | $k$=4 | $k$=8 | $k$=16 | $k$=1 | $k$=2 | $k$=4 | $k$=8 | $k$=16 |
| MoCo v2* | ImageNet | 46.3 | 58.4 | 64.9 | 72.5 | 76.1 | 11.9 | 17.0 | 22.6 | 28.1 | 32.4 |
| PCL v2* | ImageNet | 47.9 | 59.6 | 66.2 | 74.5 | 78.3 | 12.5 | 17.5 | 23.2 | 28.1 | 32.3 |
| CE (Sup.) | ImageNet | 54.3 | 67.8 | 73.9 | 79.6 | 82.3 | 14.9 | 21.0 | 26.9 | 32.1 | 36.0 |
| CE | WebVision-V0.5 | 49.8 | 63.9 | 69.9 | 76.1 | 79.2 | 13.5 | 19.3 | 24.7 | 29.5 | 33.8 |
| MoPro (ours) | | 54.3 | 67.8 | 73.5 | 79.2 | 81.8 | 15.0 | 21.2 | 26.6 | 31.8 | 36.0 |
| CE | WebVision-V1.0 | 54.5 | 67.1 | 72.8 | 78.4 | 81.4 | 15.1 | 21.5 | 27.2 | 32.1 | 36.4 |
| MoPro (ours) | | 59.5 | 71.3 | 76.5 | 81.4 | 83.7 | 16.9 | 23.2 | 29.2 | 34.5 | 38.7 |
| CE | WebVision-V2.0 | 63.0 | 73.8 | 78.7 | 83.0 | 85.4 | 21.8 | 28.6 | 35.1 | 40.0 | 43.6 |
| MoPro (ours) | | **64.8** | **74.8** | **79.9** | **83.9** | **86.1** | **22.2** | **29.2** | **35.6** | **40.9** | **44.4** |

Table 2: **Low-shot image classification** on VOC07 and Places205 using linear SVMs trained on fixed representations. We vary the number of labeled examples per-class ($k$), and report the average mAP (for VOC) and accuracy (for Places) across 5 independent runs. WebVision-V0.5 has the same number of training samples as ImageNet. The self-supervised learning methods* are trained for 200 epochs, while other methods are trained for 90 epochs. MoPro outperforms vanilla CE pretrained on Web datasets, as well as self-supervised learning and supervised learning methods pretrained on ImageNet.

across 5 independent runs. Table 2 shows the results. When pretrained on weakly-labeled datasets, MoPro consistently outperforms the vanilla CE method. The improvement of MoPro becomes less significant when the number of web images increases from 2.4m to 16m, suggesting that increasing dataset size is a viable solution to combat noise.

When compared with ImageNet pretrained models, MoPro substantially outperforms self-supervised learning (MoCo v2 (Chen et al., 2020b) and PCL v2 (Li et al., 2020b)), and achieves comparable performance with supervised learning when the same amount of web images (*i.e.* WebVision-V0.5) is used. Our results for the first time show that weakly-supervised representation learning can be as powerful as supervised representation learning under the same data and computation budget.

## 5.2 Low-resource transfer with finetuning

Next, we perform experiment to evaluate whether the pretrained model provides a good basis for finetuning when the downstream task has limited training data. Following the setup by Chen et al. (2020a), we finetune the pretrained model on $1\%$ or $10\%$ of ImageNet training samples. Table 3 shows the results. MoPro consistently outperforms CE when pretrained on Web datasets. Compared to self-supervised learning methods pretrained on ImageNet, weakly-supervised learning achieves significantly better performance with fewer number of epochs.

Surprisingly, pretraining on the larger WebVision-V2 leads to worse performance compared to V0.5 and V1.0. This is because WebVision-V0.5 and V1.0 contain the same 1k class as ImageNet, whereas V2 also contains 4k extra classes. Hence, the representations learned from V2 are less task-specific and more difficult to adapt to ImageNet, especially with only $1\%$ of samples for fine-tuning. This suggests that if the classes for a downstream task are known a priori, it is more effective to curate a task-specific weakly-labeled dataset with the same classes.

| | Pretrain Method | Pretrain dataset | #Pretrain epochs | Top-1 1% | 10% | Top-5 1% | 10% |
|---|---|---|---|---|---|---|---|
| Random init. | None | None | None | 25.4 | 56.4 | 48.4 | 80.4 |
| Self-supervised | PCL SimCLR BYOL SwAV | ImageNet | 200 1000 1000 800 | 48.8 48.3 53.2 53.9 | 62.9 65.6 68.8 70.2 | 75.3 75.5 78.4 78.5 | 85.6 87.8 89.0 89.9 |
| Weakly-supervised | CE MoPro (ours) | WebVision-V0.5 | 90 | 65.9 69.3 | 72.4 73.3 | 87.0 89.1 | 90.9 91.7 |
| | CE MoPro (ours) | WebVision-V1.0 | 90 | 67.6 **71.2** | 73.5 **74.8** | 88.3 **90.5** | 91.7 **92.4** |
| | CE MoPro (ours) | WebVision-V2.0 | 90 | 62.1 65.3 | 72.9 73.7 | 86.9 88.2 | 91.4 92.1 |

Table 3: **Low-resource finetuning** on ImageNet. A pretrained model is finetuned with 1% or 10% of ImageNet training data. Weakly-supervised learning with MoPro substantially outperforms self-supervised learning methods: PCL (Li et al., 2020b), SimCLR (Chen et al., 2020a), BYOL (Grill et al., 2020), and SwAV (Caron et al., 2020). Result for random init. is from Zhai et al. (2019).

## 5.3 Object detection and instance segmentation

We further transfer the pretrained model to object detection and instance segmentation tasks on COCO (Lin et al., 2014). Following the setup by He et al. (2019), we use the pretrained ResNet-50 as the backbone for a Mask-RCNN (He et al., 2017) with FPN (Lin et al., 2017). We finetune all layers end-to-end, including BN. The schedule is the default $1\times$ or $2\times$ in Girshick et al. (2018) Table 4 shows the results. Weakly-supervised learning with MoPro outperforms both supervised learning on ImageNet and self-supervised learning on one billion Instagram images.

## 5.4 Robustness

It has been shown that deep models trained on ImageNet lack robustness to out-of-distribution samples, often falsely producing over-confident predictions. Hendricks *et al.* have curated two benchmark datasets to evaluate models' robustness to real-world distribution variation: (1) ImageNet-R (Hendrycks et al., 2020) which contains various artistic renditions of object classes from the

| Method | Pretrain dataset | $AP^{bb}$ | $AP^{bb}_{50}$ | $AP^{bb}_{75}$ | $AP^{mk}$ | $AP^{mk}_{50}$ | $AP^{mk}_{75}$ |
|---|---|---|---|---|---|---|---|
| random | None | 31.0 | 49.5 | 33.2 | 28.5 | 46.8 | 30.4 |
| CE (Sup.) | ImageNet | 38.9 | 59.6 | 42.7 | 35.4 | 56.5 | 38.1 |
| MoCo | Instagram-1B | 38.9 | 59.4 | 42.3 | 35.4 | 56.5 | 37.9 |
| CE | WebVision-V1.0 | 39.2 | 60.0 | 42.9 | 35.6 | 56.8 | 38.0 |
| MoPro | | 39.7 (+0.8) | 60.9 (+1.3) | 43.1 (+0.4) | 36.1 (+0.7) | 57.5 (+1.0) | 38.6 (+0.5) |
| MoPro | WebVision-V2.0 | **40.7** (+1.8) | **61.7** (+2.1) | **44.5** (+1.8) | **36.8** (+1.4) | **58.4** (+1.9) | **39.6** (+1.5) |

(a) $1\times$ schedule

| Method | Pretrain dataset | $AP^{bb}$ | $AP^{bb}_{50}$ | $AP^{bb}_{75}$ | $AP^{mk}$ | $AP^{mk}_{50}$ | $AP^{mk}_{75}$ |
|---|---|---|---|---|---|---|---|
| random | None | 36.7 | 56.7 | 40.0 | 33.7 | 53.8 | 35.9 |
| CE (Sup.) | ImageNet | 40.6 | 61.3 | 44.4 | 36.8 | 58.1 | 39.5 |
| MoCo | Instagram-1B | 41.1 | 61.8 | 45.1 | 37.4 | 59.1 | 40.2 |
| CE | WebVision-V1.0 | 40.9 | 61.6 | 44.7 | 37.2 | 58.7 | 40.1 |
| MoPro | | 41.2 (+0.6) | 62.2 (+0.9) | 45.0 (+0.6) | 37.4 (+0.6) | 58.9 (+0.8) | 40.3 (+0.8) |
| MoPro | WebVision-V2.0 | **41.8** (+1.2) | **62.6** (+1.3) | **45.6** (+1.2) | **37.8** (+1.0) | **59.5** (+1.4) | **40.6** (+1.1) |

(a) $2\times$ schedule

Table 4: **Object detection and instance segmentation** using Mask-RCNN with R50-FPN fine-tuned on COCO `train2017`. We evaluate bounding-box AP ($AP^{bb}$) and mask AP ($AP^{mk}$) on `val2017`. Weakly-supervised learning with MoPro outperforms both supervised learning on ImageNet and self-supervised learning (MoCo (He et al., 2019)) on one billion Instagram images.

| Method | Pretrain dataset | ImageNet-R | | ImageNet-A | |
|---|---|---|---|---|---|
| | | Accuracy ($\uparrow$) | Calib. Error ($\downarrow$) | Accuracy ($\uparrow$) | Calib. Error ($\downarrow$) |
| CE (Sup.) | ImageNet | 36.14 | 19.66 | 0.03 | 62.50 |
| CE | WebVision-V1.0 | 49.56 | 10.05 | 10.24 | 37.84 |
| MoPro | | **54.87** | **5.73** | **11.93** | **35.85** |

Table 5: **Evaluation of model robustness** on images with artistic and natural distribution shifts. Weakly supervised learning with MoPro leads to a more robust and well-calibrated model.

original ImageNet dataset, and (2) ImageNet-A (Hendrycks et al., 2019) which contains natural images where ImageNet-pretrained models consistently fail due to variations in background elements, color, or texture. Both datasets contain 200 classes, a subset of ImageNet's 1,000 classes.

We evaluate weakly-supervised trained models on these two robustness benchmarks. We report both accuracy and the $\ell_2$ calibration error (Kumar et al., 2019). The calibration error measures the misalignment between a model's confidence and its accuracy. Concretely, a well-calibrated classifier which give examples 80% confidence should be correct 80% of the time. Results are shown in Table 5. Webly-supervised learning show significantly higher accuracy and lower calibration error. The robustness to distribution shift could come from the higher diversity of samples in Web images. Compared to vanilla CE, MoPro further improves the model's robustness on both datasets. Note that we made sure that the training data of WebVision does not overlap with the test data.

## 6 ABLATION STUDY

We perform ablation study to verify the effectiveness of three important components in MoPro: (1) prototypical contrastive loss $\mathcal{L}_{pro}$, (2) instance contrastive loss $\mathcal{L}_{ins}$, (3) prototypical similarity $s_i$ used for noise correction (equation 4). We choose low-resource finetuning on 1% of ImageNet training data as the benchmark, and report the top-1 accuracy for models pretrained on WebVision-V0.5. As shown in Table 6, all of the three components contribute to the efficacy of MoPro.

## 7 CONCLUSION

This paper introduces a new contrastive learning framework for webly-supervised representation learning. We propose momentum prototypes, a simple component that is effective in label noise

| | MoPro | w/o $\mathcal{L}_{\text{pro}}$ | w/o $\mathcal{L}_{\text{inst}}$ | w/o $\boldsymbol{s}_i$ (i.e. $\alpha = 1$) | w/o $\mathcal{L}_{\text{pro}}$ & $\mathcal{L}_{\text{inst}}$ & $\boldsymbol{s}_i$ | CE |
|---|---|---|---|---|---|---|
| ImageNet acc. | 69.3 | 68.0 | 68.2 | 68.4 | 66.9 | 65.9 |

Table 6: Ablation study where different components are removed from MoPro. Models are pre-trained on WebVision-V0.5 and finetuned on 1% of ImageNet data.

correction, OOD sample removal, and representation learning. MoPro achieves state-of-the-art performance on the upstream task of learning from real-world noisy data, and superior representation learning performance on multiple down-stream tasks. Webly-supervised learning with MoPro does not require the expensive annotation cost in supervised learning, nor the huge computation budget in self-supervised learning. For future work, MoPro could be extended to utilize other sources of free Web data, such as weakly-labeled videos, for representation learning in other domains.

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

## APPENDIX A    NOISY SAMPLE VISUALIZATION

In Figure 3, we show example images randomly chosen from the out-of-distribution samples filtered out by our method. In Figure 4, we show random examples where their pseudo-labels are different from the original training labels. By visual examination, we observe that our method can remove OOD samples and correct noisy labels at a high success rate.

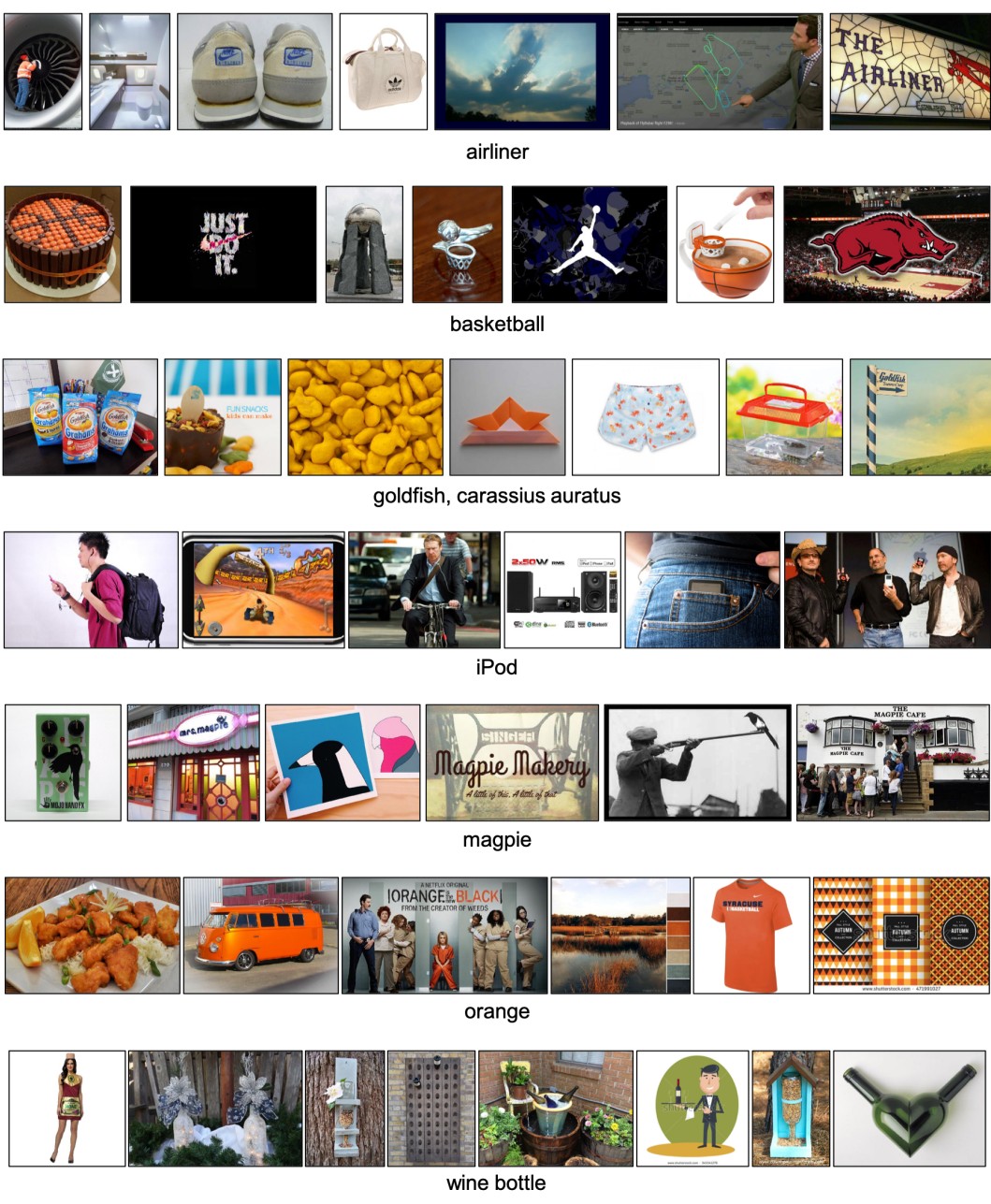

Figure 3: Examples of randomly selected out-of-distribution samples filtered out by our method. The original training labels are shown below the images.

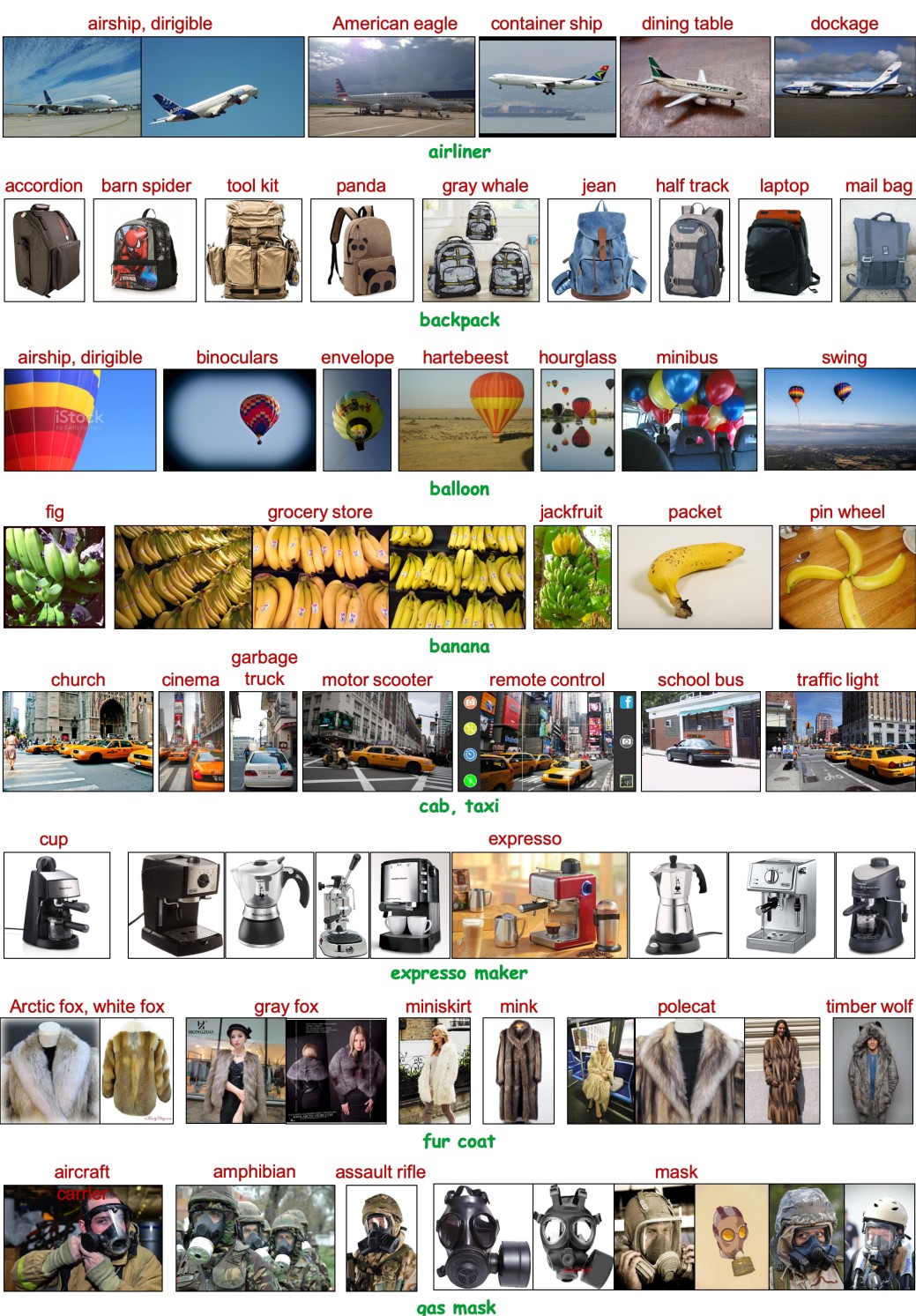

Figure 4: Examples of randomly selected samples with noisy labels corrected by our method. The original training labels are shown in red and corrected pseudo-labels are shown in green.

## APPENDIX B  PSEUDO-CODE OF MOPRO

Algorithm 1 summarizes the proposed method.

---

**Algorithm 1:** MoPro's main algorithm.

1 **Input:** number of classes $K$, temperature $\tau$, threshold $T$, momentum $m$, encoder network $f(\cdot)$, projection network $g(\cdot)$, classifier $h(\cdot)$, momentum encoder $g'(f'(\cdot))$.

2 **for** $\{(\boldsymbol{x}_i, y_i)\}_{i=1}^{b}$ **in** loader **do**   // load a minibatch of noisy training data

3   **for** $i \in \{1, ..., b\}$ **do**

4    $\tilde{\boldsymbol{x}}_i = \text{weak\_aug}(\boldsymbol{x}_i)$   // weak augmentation

5    $\tilde{\boldsymbol{x}}_i' = \text{strong\_aug}(\boldsymbol{x}_i)$   // strong augmentation

6    $\boldsymbol{v}_i = f(\tilde{\boldsymbol{x}}_i)$   // representation

7    $\boldsymbol{z}_i = g(\boldsymbol{v}_i)$   // normalized low-dimensional embedding

8    $\boldsymbol{z}_i = g'(f'(\tilde{\boldsymbol{x}}_i'))$   // momentum embedding

9    $\boldsymbol{p}_i = h(\boldsymbol{v}_i)$   // class prediction

10    $\boldsymbol{s}_i = \{s_i^k\}_{k=1}^{K}, s_i^k = \frac{\exp(\boldsymbol{z}_i \cdot \boldsymbol{c}_k / \tau)}{\sum_{k=1}^{K} \exp(\boldsymbol{z}_i \cdot \boldsymbol{c}_k / \tau)}$   // prototypical score

   // noise correction

11    $\boldsymbol{q}_i = (\boldsymbol{p}_i + \boldsymbol{s}_i)/2$   // soft pseudo-label

12    **if** $\max_k q_i^k > T$ **then**

13     $\hat{y}_i = \arg\max_k q_i^k$

14    **else if** $q_i^{y_i} > 1/K$ **then**

15     $\hat{y}_i = y_i$

16    **else**

17     $\hat{y}_i = \text{OOD}$

18    **end**

   // calculate losses

19    $\mathcal{L}_{\text{ins}}^i = -\log \frac{\exp(\boldsymbol{z}_i \cdot \boldsymbol{z}_i' / \tau)}{\sum_{r=0}^{R} \exp(\boldsymbol{z}_i \cdot \boldsymbol{z}_r' / \tau)}$   // instance contrastive loss

20    **if** $\hat{y}_i$ is not OOD **then**

21     $\mathcal{L}_{\text{pro}}^i = -\log \frac{\exp(\boldsymbol{z}_i \cdot \boldsymbol{c}_{\hat{y}_i} / \tau)}{\sum_{k=1}^{K} \exp(\boldsymbol{z}_i \cdot \boldsymbol{c}_k / \tau)}$   // prototypical contrastive loss

22     $\mathcal{L}_{\text{ce}}^i = -\log(p_i^{\hat{y}_i})$   // cross entropy loss

23    **else**

24     $\mathcal{L}_{\text{pro}}^i = \mathcal{L}_{\text{ce}}^i = 0$

25    **end**

   // update momentum prototypes

26    $\boldsymbol{c}_{\hat{y}_i} \leftarrow \text{Normalize}(m\boldsymbol{c}_{\hat{y}_i} + (1 - m)\boldsymbol{z}_i)$

27   **end**

28   $\mathcal{L} = \sum_{i=1}^{b}(\mathcal{L}_{\text{ce}}^i + \mathcal{L}_{\text{pro}}^i + \mathcal{L}_{\text{ins}}^i)$   // total loss

29   update networks $f, g, h$ to minimize $\mathcal{L}$.

30 **end**

---

## APPENDIX C  TRANSFER LEARNING IMPLEMENTATION DETAILS

For low-shot image classification on Places and VOC, we follow the procedure in Li et al. (2020b) and train linear SVMs on the global average pooling features of ResNet-50. We preprocess all images by resizing to 256 pixels along the shorter side and taking a $224 \times 224$ center crop. The SVMs are implemented in the LIBLINEAR (Fan et al., 2008) package.

For low-resource finetuning on ImageNet, we adopt different finetuning strategy for different versions of WebVision pretrained models. For WebVision V0.5 and V1.0, since they contain the same 1000 classes as ImageNet, we finetune the entire model including the classification layer. We train with SGD, using a batch size of 256, a momentum of 0.9, a weight decay of 0, and a learning rate of 0.005. We train for 40 epochs, and drop the learning rate by 0.2 at 15 and 30 epochs. For WebVision 2.0, since it contains 5000 classes, we randomly initialize a new classification layer with 1000 output

dimension, and finetune the model end-to-end. We train for 50 epochs, using a learning rate of 0.01, which is dropped by 0.1 at 20 and 40 epochs.

For object detection and instance segmentation on COCO, we adopt the same setup in MoCo (He et al., 2019), using Detectron2 (Girshick et al., 2018) codebase. The image scale is in [640, 800] pixels during training and is 800 at inference. We fine-tune all layers end-to-end. We finetune on the train2017 set (∼118k images) and evaluate on val2017.

## APPENDIX D  STANDARD DEVIATION FOR LOW-SHOT CLASSIFICATION

Table 7 reports the standard deviation for the low-shot image classification experiment in Section 5.1.

| Method | Pretrain dataset | VOC07 | | | | Places205 | | | |
|---|---|---|---|---|---|---|---|---|---|
| | | $k$=1 | $k$=2 | $k$=4 | $k$=8 | $k$=1 | $k$=2 | $k$=4 | $k$=8 |
| CE (Sup.) | ImageNet | 54.3±4.8 | 67.8±4.4 | 73.9±0.9 | 79.6±0.8 | 14.9±1.3 | 21.0±0.3 | 26.9±0.6 | 32.1±0.4 |
| MoPro | WebVision-V1.0 | 59.5±5.2 | 71.3±2.2 | 76.5±1.1 | 81.4±0.6 | 16.9±1.3 | 23.2±0.3 | 29.2±0.6 | 34.5±0.3 |
| MoPro | WebVision-V2.0 | 64.8±6.7 | 74.8±2.6 | 79.9±1.4 | 83.9±1.0 | 22.2±1.3 | 29.2±0.5 | 35.6±0.7 | 40.9±0.3 |

Table 7: Low-shot image classification experiments. Mean and standard deviation are calculated across 5 runs.

