# OpenReview forum: "MoPro: Webly Supervised Learning with Momentum Prototypes"
_ICLR.cc/2021/Conference — ICLR 2021 Poster_

### Official Review · AnonReviewer1 · 2020-10-26
**Application of recent contrastive learning methods to web images and shows improvements**

**Rating:** 7
**Confidence:** 4

**Review:**

The paper proposes to modify the recent proposed prototypical contrastive learning work (which is in turn based on improved momentum contrastive learning work) and apply it to noisy web images with detailed insights on how to combat noise. It is shown that the representation learned in this way not only achieves good results on classifying web images, but also shows great transfer performance.

+ I think it is a nice application of the recent unsupervised learning methods to settings that are more realistic (it is not practical to assume that there is absolutely no chance of getting a single label, but instead some label with noise). The proposed approach is reasonable, the motivation is strong, and the solution is (quite) simple and effective.
+ Quite a nice set of experiments are done, expanding not only the within the webvision benchmark, but also to other down-stream tasks like COCO object detection.
+ I like the robustness study showing that noisy images are good to train representations that generalize beyond normal ImageNet, but to other corrupted versions.

- In Table 1, it mentions that CE is already achieving really good results and beats most of the state-of-the-art. What might be the reason? Is it possible that stronger data augmentations are used? I am asking because most of the contrastive method these days are NOT using the same set of basic augmentations that ImageNet supervised methods are using. This makes the comparison not too fair to other methods. If possible, please consider report experiments that follow the same augmentation protocol of other compared methods.
- I noticed that different augmentations are used for the base encoder and the momentum encoder (Sec 4.2), why this is the case?
- Missing reference: (the first work that explores representation learning with web images) Chen, Xinlei, and Abhinav Gupta. "Webly supervised learning of convolutional networks." Proceedings of the IEEE International Conference on Computer Vision. 2015.
(the work that invented the word "webly"): Divvala, Santosh K., Ali Farhadi, and Carlos Guestrin. "Learning everything about anything: Webly-supervised visual concept learning." Proceedings of the IEEE Conference on Computer Vision and Pattern Recognition. 2014.
Both work actually explored some mechanism/design to de-noise web images.

---

> ### Author Response · Authors · 2020-11-13
> **Response to Reviewer #1**
>
> We appreciate the reviewer’s positive comments and valuable feedback. Here are our responses regarding the reviewers comments.
> - What is the reason for the strong performance of CE in Table 1? Is it possible that stronger data augmentations are used?
> \
> Response: Our CE approach does not use stronger augmentation and adopts the standard augmentation. There are two main reasons for it’s improved performance. The first reason is the use of class-balanced data sampling which addresses the imbalanced class distribution in WebVision. The second reason is the proposed decoupled training strategy, which consists of representation learning and classifier re-training. We hypothesize that noise can be mitigated when the classifier is re-trained on fixed representations. Performance of CE drops by ~1% without the decoupled strategy, and further by ~1% without class-balanced sampling.
> - Why different augmentations are used for the base encoder and the momentum encoder (Sec 4.2)?
> \
> Response: Different augmentations are used so that the model can learn discriminative representations that are invariant to low-level image transformations. There are two reasons why the encoder uses the standard weak augmentation. First, since only the encoder is trained with gradient update, training it using standard augmentation keeps a fair comparison with prior methods. Second, because the encoder’s output is used to update the prototypes and produce the pseudo-labels, using strong augmentations may distort the image and lead to inaccurate prototypes and pseudo-labels.
> - Missing reference: we appreciate the reviewer’s suggestion and have included these works in our paper.

---

### Official Review · AnonReviewer3 · 2020-10-27
**Anonymous review**

**Rating:** 6
**Confidence:** 4

**Review:**

This paper proposes a weakly-supervised learning method based on the self-supervised contrastive loss to combine supervised learning and unsupervised contrastive learning. A label correction mechanism is proposed by utilizing distance in class prototypes. The overall combination is reasonable and convincing.

Prons:

- As far as I know, this is the first work that combines weakly supervised learning and self-supervised learning.
- The performance on Webvision dataset is nice
- Writing it high-quality and easy to read.

Cons:

- Novelty: This method is greatly relying on an previous method, prototypical contrastive learning (PCL).  The key technical contribution is Eq (4) and (5), however, they are relatively naive and straightforward design choices.
- Experiments are a bit distracting and insufficient. I am not convinced by the larger amount of experiment focusing on tasks not actually related to weakly supervised learning (including detection, instance segmentation).
- Since it is a weakly-supervised learning method, only one table evaluates this task.
- Compared with unsupervised learning methods on downstreaming tasks are not that meaningful from my view, because the method accesses weak labels. Moreover, the noisy ratio of webvision dataset isn't that high, which can be inferred by the results in Table 6 (w/o L_pro & L_inst).
- What is the motivation of validating the methods on so many downstream tasks?
- The method replies on a temperature T, which is tuned based on datasets. The paper uses different values for webvision versions. Can it be generalized in practice?
- More weak labeled datasets more meaningful, such as Clothing1M, Food101, and real world datasets presented by this paper "Beyond Synthetic Noise: Deep Learning on Controlled Noisy Labels" (ICML2020)
- Distilling Effective Supervision from Severe Label Noise (CVPR2020) shows better Webvision results. So the SoTA claim in the abstract might not be less accurate. Proper discussions of related papers are encouraged.

---

> ### Author Response · Authors · 2020-11-13
> **Response to Reviewer #3**
>
> We appreciate the reviewer’s valuable feedback. Here are our responses regarding the reviewers comments.
> - The noisy ratio of Webvision dataset isn't that high, which can be inferred by the results in Table 6 (w/o $L_{pro}$&$L_{inst}$).
> \
> Response: The original WebVision paper (Li et al., 2017) reports 34% percent of samples as noisy. In Table 6, our result for “w/o $L_{pro}$&$L_{inst}$” still uses the classifier’s prediction for noise correction. Vanilla training with cross-entropy results in an accuracy of 65.9, which is 3.4% lower than MoPro. We have updated Table 6 to make this clearer.
> - Distilling Effective Supervision from Severe Label Noise (CVPR2020) shows better Webvision results.
> \
> Response: We thank the reviewer for suggesting this paper and have added the comparison in our paper. However, their method achieves lower accuracy than MoPro. Their paper reports a top-1 accuracy of 65.8% on ImageNet, when trained using ResNet50 and WebVision-1.0. MoPro achieves a higher accuracy of 67.8%. Furthermore, they utilize a small set of clean data from ImageNet in training, whereas MoPro does not need access to any clean data.
> - More weakly-labeled datasets are more meaningful.
> \
> Response: WebVision-V1.0 contains a diverse set of natural images from 1000 classes, whereas other datasets (e.g.  Food101 and Clothing1M) focus on a specific domain of images with fewer classes. In our paper, we aim to learn transferrable representations from noisy web images. Therefore, our experiments focus on the large-scale WebVision dataset. We respect the suggestion from the reviewer and will try to extend MoPro to other weakly-labeled datasets.
> - What is the motivation of validating the methods on so many downstream tasks and comparing with unsupervised learning methods?
> \
> Response: We aim to demonstrate that MoPro not only achieves SoTA results on classifying web images, but also learns high-quality representations that can be transferred to a variety of downstream tasks. Representation learning has high practical value and has been dominated by recent developments in self-supervised learning. However, we show that weakly-supervised representation learning with MoPro can be much more effective and efficient than self-supervised representation learning
> - The method relies on a temperature T, which is tuned based on datasets. The paper uses different values for webvision versions. Can it be generalized in practice?
> \
> Response: We adjusted the pseudo-label threshold T based on the number of classes. WebVision-2.0 uses a smaller T=0.6 because it expands the number of classes from 1k to 5k. We have experimented with using the same T=0.8 for WebVision-2.0, which only results in a slight degrade (<0.5%) in performance. Note that all the other hyper-parameters are fixed across datasets.
>
> Thanks again for your review. Please let us know if we have addressed your concerns or if you have other questions.

---

### Official Review · AnonReviewer2 · 2020-10-28
**Simple method with solid results**

**Rating:** 7
**Confidence:** 4

**Review:**

Caveat: I am not very familiar with the weak supervision literature.

Pros:
+ The method is fairly simple, a combination of three loss terms which have all been explored pretty comprehensively. The fact that a simple combination gives such solid performance improvements is a novel contribution.
+ The results are convincing and significant. I especially appreciate the low-shot transfer results: I think they are the true test of a trained feature representation.
+ The ablation and robustness studies are appreciated.

Cons:
- While experimentally the paper is on solid ground, there isn't much intuition presented as to preccisely why this combination of losses is the right thing. For example, the technique proposed in section 3.3 can also be seen as trying to use the consensus between two classifiers, one based on prototypes and the other parametric. Is this the right intuition? If so, does one need to use a prototype based classifier? Or would a nearest neighbor work fine?
- This paper inherits some of the mysteries of self supervised techniques: what function does the projector provide? what other architectural choices are important?
- I am not that familiar with the WebVision dataset, but this dataset has been collected by using Google Image Search ccirca 2017: Google might have been using internal convnets to rank images by that point, in which case it is possible that all we are doing here is replicating/slightly improving their internal network. A discussion of this would be good.


In general, I think this is a good paper, but would still like to see the authors' response to the points above.

---

> ### Author Response · Authors · 2020-11-13
> **Response to Reviewer #2**
>
> We appreciate the reviewer’s valuable feedback and positive comments on our paper. Here are our responses to the questions above.
> - The intuition of the two contrastive losses is explained as follows. The instance contrastive loss helps learn representations that are invariant to low-level augmentations and discriminative of OOD samples. The prototypical contrastive loss injects class-structural information into the representation space by clustering samples from the same class around the corresponding prototype. Together, the two losses shape a robust representation space.
> - The reviewer has the correct intuition on our noise correction method. The advantage of a prototype-based classifier over a nearest-neighbor classifier is a significant improvement in computation efficiency. The proposed momentum prototypes are updated on-the-fly without much extra computation cost, whereas a nearest-neighbor classifier requires storing the features for all training samples and performing an expensive search for each training sample.
> - The projector provides a non-linear transformation. It is a MLP with one hidden layer. Similar to SimCLR, we find that using a non-linear MLP yields better results than a linear projector. For simplicity, we follow the standard setting for our model architectures and training procedures.
> - We appreciate the reviewer’s suggestion. We are also unaware of the networks used in Google Image Search. However, the original WebVision paper reports 34% of images as noisy samples, hence it is important that MoPro addresses the noise to improve weakly-supervised representation learning.

---

### Official Review · AnonReviewer4 · 2020-10-29
**A simple and clean method with weak experimental results**

**Rating:** 6
**Confidence:** 4

**Review:**

To train a model with a noisy weakly supervised training set, this paper proposed a momentum prototypes method for label noise correction and OOD sample removal. Noise correction is done by a heuristic rule, that if the prediction is confident enough or the prediction on original label is higher than uniform probability, the label will be kept otherwise it is considered as OOD sample. For training the model, this paper jointly optimizes cross entropy loss on the corrected labels, as well as contrastive loss using prototypical examples and instances.

Extensive experiments are performed on diverse tasks, including classification, transfer, robustness, detection and segmentation.

My main concern about this paper is on the results. MoPro is a method proposed to handle the noisy training labels, while only the cross entropy baselines are compared. When I checked a few publications, and found that the reported results are significantly lower than the existing methods. For example, learning from noisy labels paper DivideMix [1] reports 77.32% top-1 accuracy  on WebVision and 75.20% top-1 accuracy on ImageNet. Based on the results, it's unclear to me why the proposed method is superior to the existing methods that learn from noisy labels.
[1] DivideMix: Learning with Noisy Labels as Semi-supervised Learning.

In Table 3, comparison to self-supervised learning is OK but not very meaningful. It would be nice to include other weakly-supervised learning results [Mahajan 2018, Kolesnikov 2020] for fair comparison purposes.

=====================

Post Rebuttal: I would like to thank the authors for the new results on WebVision-mini and ImageNet-mini, this has partially addressed my concerns as Reviewer3 raised similar issues on the SoTA claim. Overall, I think this paper is well presented and the results are solid, thus updated my rating to reflect this.

---

> ### Author Response · Authors · 2020-11-13
> **Response to Reviewer #4: MoPro achieves SoTA performance**
>
> We appreciate the reviewer’s valuable feedback. In Table 1, we have already compared with existing SoTA methods that learn from noisy labels, where MoPro achieves superior performance on the full WebVision-1.0 dataset. Regarding performance comparison with DivideMix, we would like to highlight that the reported results in DivideMix are on a mini-version of WebVision and ImageNet which only contain 50 classes. Following the reviewer’s suggestion, we also evaluated the performance of MoPro on WebVision-mini. MoPro achieves state-of-the-art performance with 77.59% and 76.31% top-1 accuracy on WebVision-mini and ImageNet-mini, respectively. Also note that DivideMix co-trains two models and uses model ensemble during inference, thus is more computationally expensive than our MoPro.
>
> Previous works on weakly-supervised representation learning [e.g., Mahajan 2018, Kolesnikov 2020] adopt a vanilla supervised learning strategy (i.e. standard cross-entropy loss) without addressing label noise and OOD noise, thus the CE results in Table 3 can represent these methods.
> \
> Self-supervised representation learning has become a popular research area for reducing human annotation cost. However, our comparison in Table 3 shows that weakly-supervised representation learning with MoPro shares the same advantage of zero human annotation, with two extra advantages: (1) substantially stronger performance and (2) much lower computational cost. Therefore, we argue that from a practical perspective, research in weakly-supervised representation learning is as important as self-supervised representation learning.
>
> Thanks again for your review. Please let us know if we have addressed your concerns or if you have other questions.

---

### Decision · Program_Chairs · 2021-01-07
**Final Decision**

**Decision:**

Accept (Poster)

**Comment:**

This paper provides an approach for weakly supervised learning by label noise correction and OOD sample removal. Overall, all reviewers agree paper is simple and approach makes sense. The experiments are solid with results on Webvision and ImageNet Mini (there were initial concerns but rebuttal handled some of those concerns). AC agrees with reviewers and recommends acceptance.